

# A framework for modelling the complexities of food and water security under globalisation

Brian J. Dermody[1], Murugesu Sivapalan[2], Elke Stehfest[3], Detlef P. van Vuuren[1,3], Martin J. Wassen[1], Marc F. P. Bierkens[4], Stefan C. Dekker[1]

[1] Copernicus Institute of Sustainable Development, Faculty of Geosciences, Utrecht University, the Netherlands

[2] Department of Civil and Environmental Engineering, Department of Geography and Geographic Information Science, University of Illinois at Urbana-Champaign, Urbana, IL 61801, USA

[3] PBL Netherlands Environmental Assessment Agency, The Hague, the Netherlands

[4] Department of Physical Geography, Faculty of Geosciences, Utrecht University, the Netherlands

*Correspondence to*: Brian J. Dermody (brianjdermody@gmail.com)

**Abstract**

In our globalised world, food security and water security are inextricably intertwined. Food production accounts for approximately 70% of global freshwater use, with variability in agricultural production impacting water resources and vice versa. Trade is central to determining water resource use, because when we trade food, we also trade the water embedded in the production of that food. As the world becomes more globalised and more urbanised, our dependence on trade for food and water security increases. Managing food and water security under globalisation is a complex challenge owing to the increased interdependency among regions and sectors. Given the unprecedented pressure on water resources in the 21[st] century, there is an urgent need for new models to assist in developing water management policies that capture these complexities.

We present a new framework for modelling the complexities of food and water security under globalisation. The framework sets out a method to capture agency, cross-scale socioenvironmental feedbacks and interdependency brought about by globalisation and urbanisation. The approach unifies and extends the existing fields of hydrology, Integrated Assessment Modelling and agent-based modelling. The core of our framework is a multi-agent network of city nodes and infrastructural trade links. This network captures the important role of cities as centres of food and water demand. In addition, it captures the infrastructural networks that constrain our ability to extract water resources from the environment and redistribute them to meet demand. We believe that this framework can form the basis for a new wave of models that capture cross-scale socioenvironmental feedbacks within our globalised world.

**Keywords:** water security, food security, integrated-assessment models, agent-based models, globalisation, urbanization



## 1 Introduction

Food security and water security are intimately intertwined (Liu and Savenije, 2008). Agricultural production accounts for approximately 70% of freshwater use by humans with variations in food production impacting water resources and vice versa (Hoekstra and Chapagain, 2006; Shiklomanov, 2000). Ground and surface water resources are important sources of agricultural water in many parts of the world, providing resilience against climate variability and increasing yields where water is limiting (Wada et al., 2010). However, in much of the mid-latitudes, the rate of ground and surface water abstraction for food production has become unsustainable, undermining both food and water security in the long-term term and having negative impacts on ecosystems (Dalin et al., 2017; Gleeson et al., 2012; Matson et al., 1997; Wada et al., 2012). Demand for water resources will likely increase in the coming decades as the global population is projected to grow by 2 billion by 2050, whilst diets are likely to become more water intensive (FAO, 2017; Gerbens-Leenes et al., 2010; Gerland et al., 2014; United Nations, 2015). In addition, water consumption for other functions such as production of fibres, energy production (including bio-energy), industry, domestic use and livestock is also expected to grow (Bijl et al., 2016).

The redistribution of food via trade is central to determining water resource use, because when we trade food, we also trade the water embedded in the production of that food (Hoekstra and Mekonnen, 2012). The redistribution of water resources embedded in food trade is known as virtual water trade (Allan, 1998). Virtual water trade plays an important role in food security (Dalin et al., 2015; FAO, 2015a). Although globally, freshwater resource use is estimated to be within the safe operating space (Rockström et al., 2009), there is enormous spatial heterogeneity in water resource stress (Gerten et al., 2011). Trade serves to redistribute food produced in water rich regions to water poor regions, and in this way virtual water trade plays an important role in ensuring food security, particularly in water stressed regions (Fader et al., 2013). By the same token, virtual water trade allows water poor regions to break free of their local eco-hydrological carrying capacities as they can import food from remote, water-rich locations (Dermody et al., 2014; D'Odorico et al., 2010; Lippman, 2010; Tamea et al., 2014). Virtual water trade is thus key to driving population growth and urbanisation and contributing to the increased demand for water resources around the globe (Suweis et al., 2013). In the worst of circumstances, virtual water trade can drive overexploitation of water resources and degradation of the environment in exporting regions, as seen in the Xingjiang Province in Western China (Liu et al., 2014), the Central High Plains of USA (Sophocleous, 2010) and much of the Eurasian midlatitudes (Dalin et al., 2017). The magnitude of virtual water trade is often underestimated as the most commonly used statistics on food trade only contain data on international bilateral trade flows (Dalin et al., 2017, 2012; Konar et al., 2011). In fact, almost all food consumed in the developed world reaches consumers through domestic or international trade networks (Chen, 2007; Seto and Reenberg, 2014; United Nations, 2012). As the world becomes increasingly globalised and urbanised our dependence on trade for food security increases (IFPRI, 2017; Rees and Wackernagel, 2008).

The globalisation of water resources via food trade means that water resource use around the world is thus increasingly determined by an interdependency of heterogeneous socioeconomic and environmental factors (Konar et al., 2016a). This interdependency occurs across spatial and temporal scales and presents an extremely complex challenge as water management policies can often have unintended consequences, which ultimately undermine food and water security (Meyfroidt et al., 2013; Sivapalan and Blöschl, 2015; Srinivasan et al., 2017). Until now, approaches to understanding unsustainable water use in food production have focused on the use of biophysical models to estimate the water footprint of food production or stylised models of socioenvironmental



interactions (Hoekstra and Mekonnen, 2012; Sivapalan et al., 2012; van Emmerik et al., 2014). These approaches maintain a disciplinary focus on water and have yet to capture the cross-scale and cross-sectoral challenges of food and water security (Konar et al., 2016a; Troy et al., 2015). Given the unprecedented pressure on water resources and the complexity of water management in the 21$^{st}$ century, there is an urgent need for new models to assist in developing effective and sustainable water management policies that capture the complexities of water management (Wagener et al., 2010). Developing new tools that capture heterogeneous and cross-scale environmental and socioeconomic interactions and the interdependence of these globally via trade is highly ambitious but we take an analogy from the discipline of climate modelling, which also began with stylised models of sub-components of the climate system. However, it became clear that these subcomponents were linked and thus it was necessary to capture these interdependencies using coupled global-scale climate models (Edwards, 2011, 2010). A similar approach is required for understanding how cross-scale and interdependent socioenvironmental interactions impact coupled water and food security, and bring about environmental change (J. Liu et al., 2015).

In this paper, we set out a new conceptual model framework for understanding food and water security that unifies existing modelling approaches and extends them by capturing the important role that globalisation and urbanisation play in determining water resource exploitation and redistribution (IFPRI, 2017; United Nations, 2012). It is our conviction that the framework we set out can form the basis for a new generation of socioenvironmental models that allow us to understand how environmental change emerges from complex and cross-scale socioenvironmental interactions. In section 2 we outline what we regard as key constraints within the global food system and the important dynamics that operate within these constraints and determine water resource use. In section 3 we outline existing approaches to understand water resource use related to food production and highlight how these approaches can be unified and extended to better understand food and water security. In Section 4 we outline our conceptual framework and how it can be implemented using existing models. In Section 5 we outline the potential applications of such a decision tool. Finally, we put forward recommendations for relevant scientific communities to make such a framework reality.

## 2. Water Resource Use and Virtual Water Trade within the Global Food System
### 2.1 The role of globalisation and urbanisation on water resources

Cities are key agents of change within the global food system (United Nations, 2012; United Nations Environmental Programme, 2013). Cities are centres of food and water demand and determine the location of physical infrastructural that constrain our ability to extract food and water resources from the environment and redistribute them around the globe (Barber et al., 2014; Rees and Wackernagel, 2008; United Nations Environmental Programme, 2013). 54% of the world's population are currently urbanised, consuming a disproportionately large, 75% of the world's resources (United Nations, 2012). The trend in urbanisation is set to continue, with an estimated 66% of the world's population expected to  live in urban regions by 2050 (United Nations, 2015). This rapid growth of cities is shifting food insecurity from rural to urban areas and driving change in both via food supply chains (IFPRI, 2017).

We consider cities and the infrastructural trade networks that link them as a key scale in terms of understanding water resource use within the globalised food system. Cities sit at the intersection of scales within the global food system and play a key role in facilitating cross-scale socioenvironmental feedbacks (Brenner, 1999;





Harvey, 1990). City nodes serve as gateways to link environmental resources in the hinterlands of cities to global markets via radiating infrastructural networks (Güneralp et al., 2013; United Nations Environmental Programme, 2013). In our framework, we define a hinterland as the agricultural area economically tied to a city. The importance of trade for cities is illustrated by the location of the world's megacities in delta regions where

resources can be accessed via inland rivers and globally via international shipping routes (Barredo and Demicheli, 2003). In fact, cities and trade coevolved (Friedmann and Wolff, 1982; Fujita et al., 2001). As city populations grow, infrastructural networks continue to upgrade capacity and extend further into natural systems, increasing natural resource extraction to meet growing demand (Ibisch et al., 2016; Laurance et al., 2015). The rate of this expansion is such, that is estimated that we will build more infrastructure in the coming 40 years than we have

built in the previous 4,000 years  (Khanna, 2016).

Demand for food in cities and their hinterlands is determined by population and socioeconomic conditions such as wealth, diets etc. (Broto et al., 2012; Stehfest, 2014). The production of food in the hinterland of cities is determined by environmental and socioeconomic conditions. Environmental conditions determine the ability to produce food whilst socioeconomic conditions constrain the ability to exploit those environmental

conditions for food production (Godfray et al., 2010). Socioeconomic conditions also constrain the ability of cities purchase food (Food and Agriculture Organisation of the United Nations, 2015a). How resources move between trading cities is determined by the socio-political and infrastructural networks that link them. Trade cannot occur directly between two cities unless the socio-political links exist and not at all if the infrastructural links are missing (De Benedictis and Tajoli, 2011).  Of course, physical infrastructural and socio-political networks cannot be

completely disentangled. Strengthening of socio-political links invariably leads to strengthening of infrastructural links, whilst the cost of investment in that infrastructure serves to stabilise those socio-political links (Khanna, 2016).

### 2.2 Complex dynamics of water resource use and virtual water trade in the global food system

Decisions related to food and water security must operate within the structures and constraints determined by cities and infrastructural networks (Rosegrant and Cline, 2003; United Nations, 2012). The food system is interdependent and a change in one element can have implications for food and water security throughout (J. Liu et al., 2015). Variability in food supply can be caused by environmental factors such as climate variability, disease, erosion, but equally by socioeconomic changes such as dietary preferences, liberalisation of land markets, trade

policy, economic potential for agricultural intensification etc. (Konar et al., 2016b; Lambin and Geist, 2008; Marchand et al., 2016; Puma et al., 2015). These socioeconomic and environmental processes are intertwined via socioenvironmental feedbacks (Sivapalan et al., 2012; Young et al., 2006). Framed simply; societies perceive environmental and socioeconomic conditions. They make decisions based upon those perceptions. Those decisions modify the environment and water resources, which in turn feeds back on future decisions (Fig. 1a).

Feedbacks operate on different temporal and spatial scales with large-scale and slow environmental processes such as aquifer recharge interacting with small-scale and fast human processes such as groundwater pumping from individual wells (Sivapalan and Blöschl, 2015). These small and fast interactions can bring about large scale emergent patterns such as aquifer depletion, which in turn feedback at the small scale on individual water users (Konar et al., 2016a).



Socioenvironmental feedbacks are linked globally and across scales via physical infrastructural and socioeconomic trade networks (Fig. 1b). Food producers in regions that are connected to global markets via trade networks are sensitive to these cross-scale socioenvironmental feedbacks (Eakin et al., 2009; Folke, 2006). For example, the 2010 drought in Russia and Kazakhstan, regions that produce about 11% of the world's wheat exports, led to a spike in the price of wheat on global markets (Nelson et al., 2014). Food producers in other wheat producing regions of the world that were well connected to global markets via physical and socio-political trade links were impacted by this price rise as they could sell their products to global markets at increased profit. Thus, environmental change in Russia and Kazakhstan impacted global markets which in turn affected the decision making of local food producers in other parts of the globe. Key to note here is that to understand which regions are sensitive to cross-scale socioenvironmental feedbacks, it is essential to capture the trade networks that link regions to global markets. The more connected regions are to global markets via networks, the more sensitive they are to these cross-scale socioenvironmental feedbacks (Pande and Sivapalan, 2016).

**Figure 1. Complex feedbacks within the food supply system that lead to global water use patterns.**

The networks that link producers and consumers increase the interdependence of the food system, making it challenging it challenging to develop water management policies that do not result in unintended consequences (Geist and Lambin, 2002; Hoekstra and Hung, 2005). For example, policy aimed at restricting unsustainable groundwater abstraction in agriculture in one region of the world may simply result in that production being displaced to another region with potentially weaker regulation and more serious environmental impacts than would have occurred in the original country (Konar et al., 2016a; Meyfroidt et al., 2013). Interdependence also exists across sectors with the mechanisation of agriculture leading to an increasingly tight coupling between food, water and energy (United Nations Water, 2015). Currently 30% of energy produced is used in food production, with fluctuations in energy costs having direct impacts on agriculture and thus water resources (Bazilian et al., 2011; Frieler et al., 2015).

### 2.3 Food and water security under globalisation

Food security requires a food system that is resilient to fluctuations in water availability (FAO, 2015a). Equally, long-term food security requires sustainable water resource use (Gleeson et al., 2012; Lang and Barling, 2012). Currently, food security is maintained through three main mechanisms: heterogeneity of supply chains, diversification of consumption and production and investment in food reserves (Godfray et al., 2010). Each of these strategies is followed to a greater or lesser degree in regions around the world. Food reserves have been identified as a key factor in increasing food security because they provide a buffer against variations in yields and thus reduce market volatility (Fraser et al., 2015). They may also be used to increase the sustainability of water resource use in water limited regions, because reserves arising from over-production in wet years can be used to limit water abstraction for irrigation in dry years. However, if building-up of reserves requires over-production and increased losses, reserves can contribute to increased water resource use. For water-limited regions, reserves may thus be an unsustainable solution for long-term food security. In recent years there has been a trend of countries to decrease reserves and increase imports to boost food security owing to the expense of producing and storing extra food (Marchand et al., 2016).



Trade reduces the reliance on reserves because regions can export when they have a surplus and import when they have a deficit (Dermody et al., 2014). Equally, trade has the potential to increase the efficiency with which water resources are used globally (Chapagain and Hoekstra, 2006). For example, the water required to grow a kilogram of wheat varies from place to place owing to several environmental and socioeconomic factors. Water is thus saved if countries import from regions with more efficient water-use to yield ratios than their own (de Fraiture et al., 2004; Hanasaki et al., 2010; Konar et al., 2013). However, trade enables regions to grow beyond their local land and water constraints and is central to population growth of water-poor regions such as Saudi-Arabia (Barredo and Demicheli, 2003; Curtis, 2009). In addition, increased demand in wealthy importing countries can trigger over-exploitation of water resources and degradation of the environment in exporting countries that seek to benefit from that demand (Dalin et al., 2017; Sophocleous, 2010; van Emmerik et al., 2014). Trade also exposes importing countries to environmental changes in other regions. For example, during water-stressed periods, exports from producing countries are often reduced, thus increasing volatility on a global scale (Tangermann 2014). An alternative measure of increasing resilience is for consuming regions to diversify the type of foods they consume so that fluctuations in one food commodity can be offset by importing an alternative food type. Indeed, there has been a trend of increasing diversification of consumption in recent decades but this likely emerges from changes in consumer demand rather than top-down food security policies (Kearney, 2010).

It remains an open question as to which strategy or combination of strategies provides more sustainable use of water resources and greater resilience to fluctuations in water resource availability (Haile et al., 2015; Marchand et al., 2016). Since the food system is heterogeneous and interdependent, the optimally resilient and sustainable solution will be a combination of strategies and always changing depending on what happens elsewhere. Therefore, decision tools are required that allow us to develop tailored and dynamic solutions that account for heterogeneities and interdependencies.

**2.4 Path Dependency**

Even where secure and sustainable food production and water use pathways are identified, the ability of regions or sectors to transition to those pathways depends on how path dependent they are along their current development pathways (Kay, 2003; Pierson, 2000; Strambach, 2008) (Fig. 2). Path dependency emerges from economic, infrastructural and cultural investments societies make towards some optimum. Investments that move a system towards an optimum are reinforced over time, such as investment in irrigation infrastructure that bring about increased yields and profits. The more investments made towards that optimum, the costlier it is to transition to an alternative optimum (Sydow et al., 2009). Infrastructure construction is one of the most powerful forces driving path dependency (Liebowitz and Margolis, 1995). For example, investment in water management infrastructure is theorised to have been a central force driving the cultural and economic development of the early civilisations in Mesopotamia and China (Giordano et al., 2004; Wilkinson et al., 2015). In the context of unsustainable water use, it is thus crucial to take account of the constraints that infrastructure places on regions or sectors to adapt to sustainable water use pathways.

The case of the Central High Plains of the United States provides an example of where infrastructural investment has created lock-in along an unsustainable pathway. Investment in irrigation infrastructure in Central High Plains began with the resettlement of the region following the Dust Bowl crisis of the 1920's (Brodwin, 2013). Over decades, increasing agricultural returns brought about increased investments in irrigation



infrastructure and caused the economy of the region to become dependent on water abstraction from the Ogallala Aquifer (Sophocleous, 2005; Torell et al., 1990). The rate of abstraction has accelerated in recent years and has become unsustainable in many parts of the aquifer (Sophocleous, 2010). In certain regions, the fuel costs for pumping irrigation water have begun to exceed the profits farmers gain from selling their produce. Nonetheless,

the existing practices are maintained owing to the prohibitively high costs, in the short term, of transitioning to alternative development pathways (Liebowitz and Margolis, 1995). The economy in the Central High Plains is now highly exposed to climate change as they lack the groundwater resources to dampen the impact of climate variability on yields (Sophocleous, 2010). It is essential to understand the role that infrastructural investment and path dependency play in determining the capacity of regions to transition to sustainable water use pathways.

Decision tools that capture these infrastructural constraints can be used to identify regions or sectors that are at risk from lock-in to unsustainable pathways and explore the possibility space for transitioning to sustainable water use pathways.

**Figure 2.  Path dependency and water use.**

**3. Approaches to understanding water resource use in the global food system**
       **3.1 Water Footprint Studies**
       There has been much work done to quantify water use in food production, also referred to as the Water Footprint of food production (Chapagain and Hoekstra, 2006; Konar et al., 2011). Water Footprint studies apply hydrological models to compute moisture storage in the soil as well as the water exchange between the soil and

the atmosphere and the underlying groundwater reservoirs based on specific crop water-use factors (van Beek et al., 2011; Wada et al., 2011). Using this technique, it is possible to estimate the amount of precipitation (green water) or surface and groundwater (blue water) that is used in agricultural production for a given crop cover and environmental and agronomic conditions (Mekonnen and Hoekstra, 2011). Water footprint studies show that in many parts of the world, particularly the mid-latitudes, the extraction rate of surface and groundwater resources

for agricultural production exceeds the recharge rate of surface and groundwater reservoirs (Gleeson et al., 2012; Wada et al., 2012) (fig. 3a). The implication of this is that much of the mid-latitudes will become much less agriculturally productive in the future if current unsustainable agricultural practices are continued.

            Virtual Water studies marry country-level data on food production and trade from sources such as the World Bank and FAO with water footprint analysis to estimate the volume of green and blue water that is

redistributed virtually around the globe as food and biofuels (Dalin et al., 2012; Hanasaki et al., 2010) (Fig. 3b). Virtual water studies highlight that many countries around the world are dependent on foreign water resources for their food security. For example, it is estimated that the Middle East imports more water virtually than flows down the Nile each year (Barnaby, 2009). Although the focus has primarily been on blue and green water fluxes, recent work has focused on quantifying water that is unsustainably extracted, which is particularly revealing of the

regions and sectors that are driving unsustainable water resource around the world (Dalin et al., 2017; Gleeson et al., 2012; Wada et al., 2012; Wada and Bierkens, 2014). For example, demand for cotton in the Western Hemisphere drives unsustainable water resource extraction in Southern Asia (Chapagain et al., 2006). Water footprint studies have primarily been focused at the spatial scale of countries as much of the data on trade fluxes is available at that scale. However, increasingly there are attempts to quantify virtual water flows within countries

(Dalin et al., 2014; Dang et al., 2015) and even the water footprints of cities (Hoff et al., 2014). This is a critical



step as country-level patterns are emergent from cross-scale socioenvironmental processes. In order to understand these larger-scale patterns, we must begin to understand how they emerge from the bottom-up (Konar et al., 2016a).

**Figure 3. Water footprint studies.**

### 3.2 Sociohydrological Studies

Water footprint studies quantify the rate of unsustainable water resource abstraction and how food trade relates to unsustainable water resource use. However, illuminating the complex socioenvironmental mechanisms that bring about emergent water use is not the focus. Sociohydrology provides promise in this regard. Sociohydrological

studies set out to understand emergent water use patterns by capturing cross-scale interactions in space and time between humans and water resources. In doing so, these studies illuminate the complex socioenvironmental mechanisms that may help human societies to keep water extraction within sustainable limits, drive it unsustainably towards environmental degradation and even collapse, or take timely action (following unsustainable use initially) towards eventual recovery (Sivapalan et al., 2012).

15        In the last few years there have been several sociohydrological studies that have explored the competition between water for food and water for the environment in several large agricultural basins (Sivapalan et al., 2014). They have done so by explicitly considering bi-directional feedbacks between humans and the environment in these basins, leading to different types of emergent dynamics. An example of emergent dynamics is a "pendulum swing", in that communities have alternated between water extraction for agriculture in the early stages of

development, followed by subsequent efforts to mitigate or reverse the consequent degradation of the riparian ecosystems (Kandasamy et al., 2014; Liu et al., 2014). This has been explained by counteracting productive and restorative forces, mediated via technology, environmental awareness and the intervention of governance institutions (Elshafei et al., 2015; D. Liu et al., 2015; van Emmerik et al., 2014). Another example of emergent dynamics is the "irrigation efficiency paradox", whereby agricultural water that is saved through irrigation

efficiency measures only enabled farmers to bring more land into production, and contributing to increased (not decreased) water consumption, thus wiping out any gains of using such technologies (Scott et al., 2014). The sociohydrological studies so far have explained the observed emergent phenomena by allowing human agency, the capacity of humans to make decisions to benefit themselves and/or their environment, to become endogenous to the coupled human-water system models (van Emmerik et al., 2014; Liu et al., 2015; Elshafei et al., 2015), a

key innovation in the development of sociohydrological models.

        Analyses of sociohydrological systems have so far assumed the systems of concern are isolated entities in space, e.g., an agricultural river basin. Increasingly, in a globalised world, many different such entities may be linked through trade in goods (e.g., food) that human agents present in them either produce and/or consume. One may then envisage a collection of basin-scale sociohydrology's that are tele-coupled through trade networks

(d'Amour et al., 2016). With the presence of these trade networks, the effect of changes in the global economy can cascade down to national, basin and even sub-basin scale local water management or decision-making. For example, a shock in global food prices can cascade down to the sociohydrological system for a given river basin. Conversely, the effect of local water use practice or production technology can be up-scaled to the economy of a larger sociohydrological basin, or to broader, regional economy through economic tele-connections. All of these

considerations call for an extension of the endogenization of human agency to space and to space-time, presenting



an opportunity to manage sustainability of water use and assess water security at regional and global scales (Pande and Sivapalan, 2017; Srinivasan et al., 2017).

### 3.3 Beyond a disciplinary focus on water

Water footprint and sociohydrological studies have provided and continue to provide an invaluable understanding of water use patterns in the global food system (Dalin et al., 2017; Konar et al., 2016a). However, they are somewhat limited by their disciplinary focus on water. Lambin et al. (2000) highlight that an integrated approach is important to understanding environmental change because factors such as changes in technology, energy sector, diet etc. all play an important role. Thus, it is necessary to integrate many processes within a single, unifying

framework (J. Liu et al., 2015). This has been the core focus of Integrated Assessment Models (IAMs) over several decades (Weyant et al., 1996). IAMs are influential decision-making tools to advise policy by simulating interacting socioeconomic and environmental processes at regional-global scales (IPCC, 2015). Until now, IAMs have been principally focused on understanding and projecting greenhouse gas emissions, however increasingly, complex agroeconomic modules are being included within these models (Stehfest, 2014). The advantage of an

agroeconomic model nested within an IAM is that it captures changes in agriculture linked to factors such as energy production, population growth, dietary changes etc. In developing an understanding of the complexity of water use in agriculture it is essential to capture these cross-sectoral interdependencies (Bazilian et al., 2011). IAMs, thus provide a promising pathway for capturing the cross-sectoral interdependence of water use (Bijl et al., 2016).

20       The focus of IAMs is to provide future projections of environmental change in response regional-scale exogenous drivers such as population growth, dietary change etc. A complication however, is that water use in food production is emergent from cross-scale, socioenvironmental interactions among heterogeneous entities, whether they be farmers, cities, countries etc. (see Section 2). These interactions bring about emergent higher level market changes which feedback on water resource use decision-making across scales (Konar et al., 2016a;

Sivapalan and Blöschl, 2015). These emergent processes are not currently captured in IAMs. To do so would require a coupled description of numerous relevant process at different scales. While some of this has been included in the most complex IAMs, it risks including too much complexity, possibly leading to a trade-off with transparency (van Vuuren et al., 2016).

       An alternative method for understanding emergence in socioenvironmental systems is through the

application of agent-based models (ABMs) (An, 2012). ABMs consist of multiple heterogeneous agents whose interactions with each other and their environment bring about higher-level emergent patterns. Because ABMs capture how higher-scale patterns emerge from the bottom-up, they can illuminate the pathway at different scales that lead to those higher-level patterns (Filatova et al., 2013, 2009). Thus, they can inform policy at different scales which is essential in the context water resource decision making, where cross-scale interactions in space

and time provide a particularly difficult challenge (Lach et al., 2005). ABMs have been successfully applied at a small scale to understand changes in water resources owing to food production. For example, an ABM was developed to understand the diffusion of irrigation technologies among farming communities in Chile. In this model, economic and hydrological components were coupled so researchers could capture the impact of technological adoption on the hydrological systems and associated feedbacks on the farmers (Berger, 2001).

However, they have yet to be incorporated in global models and the means to do so remains challenging (Helbing,



2013). For example, it is not yet feasible to capture individual farmers at a global scale, from a data or computational perspective. In our framework, we propose that cities pose an ideal scale in which to understand emergent food and water use patterns, because cities are key agents of change and lie at the intersection of scales within the global food system.

## 4. Modelling Framework

In the following section, we outline our framework, which seeks to capture the structure and constraints of the food system and the dynamics that operate within these constraints and bring about emergent water use patterns. The core structure of our framework is a multi-agent network of city nodes and trade links (Fig. 4). In our

framework, cities are agents and comprise of an urban area and associated hinterland. Environmental conditions in the hinterlands of city agents are based on grid-based model of the biophysical environment. Socioeconomic conditions in the hinterlands of cities are determined from an Integrated Assessment Model. Socioeconomic conditions constrain agent's ability to exploit environmental conditions within their hinterland for food production. City agents have a common utility function: to satisfy local and market demand for food. They do this via food

production and trade. How each city achieves this differs based on the heterogeneous socioeconomic and environmental conditions within their hinterland and the socioeconomic and infrastructural networks that link cities. The interactions among cities and their environment bring about higher level emergent patterns which feedback to the biophysical model and IAM in the next simulation step. In this way, small-scale interactions among heterogeneous city agents and their environment are incorporated within existing macro-scale model

structures without changing the structure or complexity of those models.

**Figure 4. Modelling Framework.**

### 4.1 Structure of the multi-agent network

Each city agent has an associated hinterland. The size of a hinterland depends on the ease of transportation between the city and the agricultural area, which is determined by geographic, infrastructural and socio-political factors (Billen et al., 2009). To capture each of these elements, we define city hinterlands based on the hierarchical overlay of supra-sub national administrative borders and theissen polygon operation among cities based on cost-distance of trade via road, rail and inland water ways (Berthelon and Freund, 2008) (Fig. 5). The hinterland is restricted to

land contiguous with the urban area and that falls within the same administrative region. For many cities, and depending on the food commodity, the effective hinterland will extend beyond these contiguous administrative regions (Billen et al., 2009; Güneralp et al., 2013). However, policy is applied at the scale of these administrative boundaries and cities operate within these policy constraints. If policies stimulate free trade between administrative regions, then the effective hinterland of a city can expand (Knox and McCarthy, 2012) as is seen

in cities such as Basel, whose effective hinterland extends across 3 national borders. Whereas with a city such as Seoul, the border with North Korea is an impermeable boundary to the hinterland owing to the lack of trade relations or infrastructure between the two (Food and Agriculture Organisation of the United Nations, 2015b). Therefore, the framework allows users to explore the impact of trade policy at different scales and infrastructural change on food and water use and virtual water flows. Hinterlands vary in terms of size and composition. For

example, the hinterland of a city in Western Australia will be large, with a high proportion of natural landcover





and low population density, whilst a hinterland in the Eastern China will be smaller with a high proportion of agricultural and urban landcover and high population density. Certain cities will have a net demand for food resources whilst others, with large agricultural hinterlands, will have a net surplus in food resources (Dermody et al., 2014).

**Figure 5. City agent properties.**

The trade network that links city agents is hierarchical. The network comprises a lower-level physical infrastructural network and an upper-level network of macro-scale bilateral trade patterns (Fig. 6). The lower level of the trade network is defined by physical trade infrastructure of roads, rail and shipping lanes (Berthelon and

Freund, 2008; Karpiarz et al., 2014; Limão and Venables, 2001). The data required to build this network, such as open street maps and database of cargo ship movements, is openly available (Haklay and Weber, 2008; Kaluza et al., 2010; Thomas Brinkhoff, 2016). Each link type has a different transport cost. For example, the cost of bulk trade is approximately 7-times less by ship than by road (Limão and Venables, 2001). In addition, intermodal transport costs are applied for transferring goods between transport modes (Janic, 2007). Infrastructure

development can be projected using infrastructure growth algorithms or manually added to explore the impacts of prospective plans such as the new trans-Eurasian Silk Road Economic Belt (Ahmed et al., 2013; Arima et al., 2008; Brugier, 2014; Walker et al., 2013). The infrastructural network constrains where resources can be extracted from the environment and redistributed to meet demand (Barber et al., 2014; Khanna, 2016). Thus, all resource flows should travel via the lower level physical infrastructural network in a realised version of the model

framework.

The upper level network constrains the trade among cities in different countries. The upper level network consists of bilateral trade links among countries. Link weights between countries are calculated using a Computable General Equilibrium (CGE) model, which is embedded in many IAMs (Stehfest et al., 2014; Wicke et al., 2015). CGEs capture historical bilateral trade balances, competitiveness (relative price developments) and

trade policies to estimate trade patterns (Woltjer et al., 2014). CGEs are often criticised for being overly reliant on data constraints and having weak econometric foundations. However, CGEs are well-suited to understanding how country-regional scale changes in an interdependent economy affect resource reallocation. Thus, they are suited to exploring the interdependency of regions via trade (Hertel et al., 2007; Hertel and Hertel, 1997). City agents within the same country trade with one another based solely on supply and demand for each food

commodity and the cost-distance among cities. Cities in different countries are also constrained by the upper-level network of bilateral trade links. If the CGE simulates high trade volumes between two countries, then agents in those two countries will also have a high probability of bilateral trade. However, it is important to underline that agent food production and trade decisions are stochastic and based on the utility function to satisfy food demand at minimum cost. Thus, for given constraints there will be a range of possible solutions. Equally, if conditions

change at the small scale, this will result in alternative emergent patterns at the large scale. The emergent trade patterns are aggregated at the country and region scale of the CGE and fed back into it for the next simulation year. In this way, cross-scale processes and agency are married with conventional general equilibrium modelling approaches.

**Figure 6. Virtual Water Trade Network.**



### 4.2 Food production and water use

Within our framework, city agent food production decisions are constrained by socioeconomic and environmental conditions within their hinterlands. The environmental conditions within the hinterlands of cities are determined using a process-based model of the biophysical environment. A range of global ecosystem, hydrological and crop models have been developed in recent years (Bierkens and van Beek, 2009; Bondeau et al., 2007; Smith et al., 2014). These models are focused to a differing extent on capturing different environmental processes such as hydrological, ecosystem or crop growth processes. In terms of capturing the impact of food production on water resources, it is essential to capture the interaction among hydrological, ecosystem and crop growth processes, including irrigation, or so-called ecohydrological processes (Bierkens, 2015). This is because natural ecosystems play an important role in the terrestrial water cycle and changes in ecosystems can impact the water cycle and water availability for agricultural production (Alcamo et al., 2005). Equally, agriculture can impact ecosystems through modification of the hydrological cycle (Kingsford and Thomas, 2004). It is thus necessary that models developed using our framework couple hydrological, ecosystem and crop models to understand the heterogeneous ecohydrological constraints on food production in the hinterlands of city agents. Each city hinterland has a potential yield and water resource usage based on these environmental constraints.

Socioeconomic conditions constrain the ability of cities to exploit environmental conditions in their hinterlands for food production. Socioeconomic determinants of food production (agroeconomic) are captured within some IAMs. For example, the IMAGE IAM contains the dedicated agroeconomic model MAGNET (Modular Applied GeNeral Equilibrium Tool) (Woltjer et al. 2014). MAGNET is a modified version of the GTAP CGE model specifically designed to capture macroscale agroeconomic processes. MAGNET calculates regionally heterogeneous agricultural intensification levels based on data related to fertilizer usage, irrigation prevalence etc. gathered by organisations such as the World Bank and FAO. These agroeconomic conditions can be calculated for 134 countries and regions and aggregated to the 26 regions used within the IMAGE framework. For example, a Western European region will have very different agroeconomic conditions compared with a sub-Saharan African region within an IAM (Stehfest et al., 2014).

Agroeconomic models also calculate per capita food demand at the regional scale based on factors such as affluence and changing diets (Stehfest, 2014). Per capita regional demand can be distributed among cities in our framework based on spatially explicit grid-scale population estimates (Klein Goldewijk et al., 2011; Thomas Brinkhoff, 2016; UN Population Division, 2015). Cities and their hinterland are either in surplus or deficit for a crop type based on (local production – local demand) (Dermody et al., 2014). Demand is also based on trade demand calculated in the trade component of the framework. Areas close to large population centres will tend to have higher export demand than those far away (Berthelon and Freund, 2008) (see section 4.3).

Cities can meet food demand through agricultural expansion, agricultural intensification or imports. The relative contribution of agricultural expansion or intensification to food production is one of the most uncertain and crucial factors in global land-use modelling (Alexander et al., 2016). It can be based on historic data or explicitly modelled via elasticities of both intensification and expansion (e.g. substitutability between fertilizer, labour, capital and land, and land supply elasticity) (Schmitz et al., 2014; Stehfest et al., 2013). For example, the IMAGE IAM contains land change algorithms that estimate the suitability for agricultural expansion or intensification based on biophysical suitability and factors such as land availability, protected areas and on changes in crop yields due to future climate change (Letourneau et al., 2012; Neumann et al., 2011; Stehfest et al.,



2014). The algorithms used in existing IAMs should eventually be extended to include detailed infrastructural data as well as more complex hydrology in determining agricultural suitability (Barber et al., 2014; Wada et al., 2012; Walker et al., 2013) (Fig. 7).

In our framework, cities with increasing demand, low (high) agricultural intensity potential and high (low) expansion potential are therefore likely to expand (intensify) agriculture (Fig. 7). Cities that are constrained from expanding or intensifying agriculture will increase imports. In fact, cities will implement each of these strategies to a greater or lesser degree. The agent decision process will generate land-use maps that are prescribed to the biophysical environment and the impact on ecohydrology is calculated (Fig. 8a) (Biemans et al., 2013; Wada et al., 2012). In this way, macroscale agroeconomic food production processes are blended with small-scale

biophysical, demographic and physical infrastructural conditions. As mentioned, a key advantage of using IAMs is that they capture interdependencies across sectors and therefore can capture changing constraints on food production based on changes in energy prices, for example (Bazilian et al., 2011).

**Figure 7. Food production decisions.**

**4.3 Virtual Water Trade**

In our framework, the resources produced in the hinterland of cities are either consumed locally or exported to meet market demand. Cities are in demand for a crop if (local production – local demand) in the city hinterland is negative and have a surplus if (local production – local demand) is positive (Dermody et al., 2014). Cities with a

demand, import from cities with a surplus based on the cost-distance among cities (Berthelon and Freund, 2008). Thus, the probability of trade among cities decays with increasing cost-distance (Karpiarz et al., 2014; Limão and Venables, 2001). As mentioned, cities can only trade with cities within the same country or cities in different countries with a probability based on the upper-level network of macro-scale bilateral trade patterns (Fig. 6). Because city agent trade decisions are stochastic, it allows for alternate emergent trade patterns within the

constraints of the same network. In this way, the solution space for given constraints can be explored (An, 2012). These emergent patterns are aggregated to the level of regions used in the CGE and provide the initial conditions for the next simulation year of the CGE (Fig. 8b). If conditions change at the city-scale, that will change the interactions among agents, leading to alternative emergent higher-level patterns. In this way, macroscale agroeconomic trade processes are blended with small-scale biophysical, population and physical infrastructural

30

conditions as well as agency. This is a crucial step in capturing cross-scale feedbacks within the global food system using existing models.

**Figure 8. Modelling Framework Workflow.**

**4.4 Flexibility and modularity**

35

The framework outlined provides a flexible and modular platform for coupling biophysical and IAM models via the multi-agent network of city nodes and trade links. The network of city nodes and trade links lies at the intersection of scales within the global food system and is a central constraint for a range of research questions related to environmental changes arising from food production and trade. The multiagent network receives socioeconomic and biophysical constraints from IAM and biophysical models in standard model output such as

40

raster NetCDF files, databases, networks (see Dermody et al., 2014). Agents make food production and trade



decisions based on those constraints and the modified data are sent back to the IAM and biophysical models as input for the next simulation step. Such flexibility and modularity will enable a wide range of research questions on unsustainable water resource use and facilitate model intercomparisons which are essential for sensitivity analyses and the transparency of the decision tool (Wicke et al., 2015). It also presents a unique method to

incorporate cross-scale processes and agency within existing macroscale model structures. This is an important step in developing current approaches to incorporate agency, cross-scale socioenvironmental feedbacks and the interdependency of these via trade (Helbing, 2013; Müller-Hansen et al., 2017; Rockström et al., 2017).

### 5. Applications of models built using the framework

Models based on the framework presented have the potential to capture the socioeconomic, environmental and infrastructural constraints on food production and trade. Within these constraints, the stochastic nature of the multi-agent model will enable the exploration of the possibility space for realizing food security requirements and the associated impacts on water resources. The emergent possibility space illuminates how constrained regions or sectors are based on socioeconomic, environmental and infrastructural factors to transition to sustainable water

use pathways (Brown et al., 2005; Garud et al., 2010). Those regions or sectors that are locked into unsustainable water use pathways should be prioritised for intervention (Fig. 2). Our framework also captures interdependencies across regions and can be used to identify optimal trade-offs between different environmental stress factors. For example, optimality algorithms can be applied to find optimal solutions from a land saving and water use (Dermody et al., 2011; Konak et al., 2006; Vrugt et al., 2003). Equally, because the IAM captures cross-sectoral

interdependencies, trade-offs or synergistic solutions can be explored from a water and energy perspective (Bazilian et al., 2011; Stehfest et al., 2013). Such synergistic approaches are key to addressing the challenges set out in the UN Sustainable Development Goals (Costanza et al., 2016; Lu et al., 2015).

The framework can be applied across scales to investigate changes at catchment, city or global scales. Given the globalised nature of water resources, it is essential that smaller scale studies can be linked with larger

scale hydrological or market changes (Pande and Sivapalan, 2016; Sivapalan and Blöschl, 2015; Verburg and Overmars, 2009). Returning to the parallels with climate models, mesoscale climate models apply largescale climate patterns as boundary conditions and close the energy balance within small scale models (Douglas et al., 2009). Equally, the model framework outlined can be used to understand how detailed changes within a city or catchment link to large-scale market or hydrological changes. Thus, models built according to our framework can

contribute and benefit from the rapid growth of within-city studies such as studies of urban metabolism where the fluxes calculated in our framework are boundary conditions or sociohydrological studies which focus on the catchment scale (Broto et al., 2012; Liu et al., 2014; van Emmerik et al., 2014; Zhang, 2013). In terms of informing policy, our approach can uncover potential cross-scale feedbacks or cascading effects of policies to other regions or sectors (Lambin and Meyfroidt, 2011). Thus, models built using the framework outlined are ideally suited to

informing policy on so-called "wicked problems" associated with water resource use (Dentoni et al., 2012; Duit and Galaz, 2008; Lach et al., 2005).

The approach also presents a pathway to expand the capabilities of IAMs. IAMs are highly influential decision tool for policy makers. IAMs exploit scenarios, such as the recently published Shared Socioeconomic Pathway (SSP), which outlines future environmental change according to different socioeconomic development

storylines (O'Neill et al., 2015). However, these scenarios are applied exogenously in IAMs and therefore struggle



to capture emergent and non-linear change typical of socioeconomic change (Rockström et al., 2017). Equally, they generally focus at the scale of supranational regions and decadal time periods. However, political decision making often has a short-term outlook and is applied at a range of temporal and spatial scales. Our approach simulates large-scale change as an emergent, bottom up process. Thus, it has the potential to illuminate the pathway at different scales that leads to large scale patterns demonstrated in IAMs. In terms of informing policy, this is an essential step as it allows policy makers to explore the impacts of policy changes at the different spatial and temporal scales at which policy is applied. In doing so, approaches such as this can help illuminate roadmaps, considering heterogeneous constraints, to reach regional and long term goals outlined in IAMs (Rockström et al., 2017). This can be a potentially important step in extending the usefulness of IAMs whilst avoiding adding unwanted complexity to those models (van Vuuren et al., 2016).

## 6. Summary and recommendations

The complex and intertwined nature of food and water security within our globalised and urbanised world requires new models and approaches to inform policy makers. The framework we present unifies and extends the existing fields of hydrology, Integrated Assessment Modelling and Agent-Based modelling to understand water use patterns that emerge via cross-scale socioenvironmental processes and the interdependence of these globally via food trade networks. It is our conviction that models built according to the framework outlined will represent a new wave of global socioenvironmental models that can provide unprecedented insights into the complexity of socioenvironmental interactions within our globalised world. Much of the groundwork for the framework already exists, however some extra efforts are required to achieve an operational model.

The main step required to make this framework a reality is the construction of a multi-agent network of city nodes and trade links. As outlined, cities and infrastructural trade networks play a key role in environmental changes as they constrain our ability to extract resources from the environment and redistribute them around the globe. Thus, a multi-agent network of cities and infrastructural links will provide a platform for a wide range of research questions related to food production and environmental change. The resources required to build this network are openly available but need to be translated to a scalable and flexible network architecture. The structure of this network should be designed in collaboration with scientists from a wide range of disciplines to ensure the relevant elements are incorporated to meet a range of research questions.

In order to facilitate collaboration in building such a multi-agent network, there is a need for interdisciplinary dialogue and collaboration. Indeed, this is essential for achieving the Sustainable Development Goals which present interdependent challenges across disciplines (Costanza et al., 2016; Lu et al., 2015). In our experience, the theories and methods associated with complexity science provide an ideal approach for tackling interdependent sustainability challenges (J. Liu et al., 2015). Complexity is not just a suite of theories and methods, it delivers an intuitive way of understanding interdependent systems and provides a platform for deep interdisciplinarity collaboration that is required to meet today's sustainability challenges.

**Competing interests**. The authors declare that they have no conflict of interest.





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





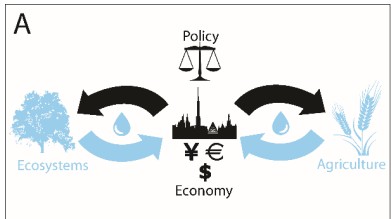

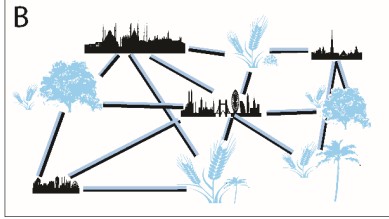

**Figure 1. Complex feedbacks within the food supply system that lead to global water use patterns.** (A) Humans perceive environmental and socioeconomic conditions. They make food production and trade decisions based upon those perceptions, guided by prevailing norms and values in society (Elshafei et al., 2016; Sivapalan et al., 2014), modifying the environment and water resources. Those changes feedback to influence decision making further. (B) These feedbacks are linked globally via the food trade network across dynamic and varied socioeconomic, political and environmental settings (Pande and Sivapalan, 2016).

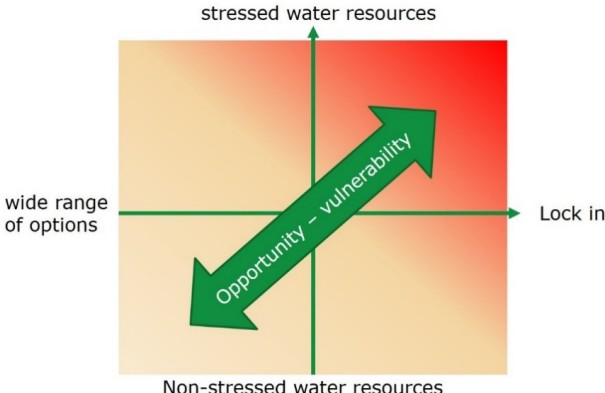

**Figure 2. Path dependency and water use.** Regions that have low levels of path dependency and non-stressed water resources have a high potential for a low-cost transition to sustainable water use pathways. Regions that are highly path dependent along an unsustainable pathway require urgent, large investments or innovations to transition to sustainable water use pathways.



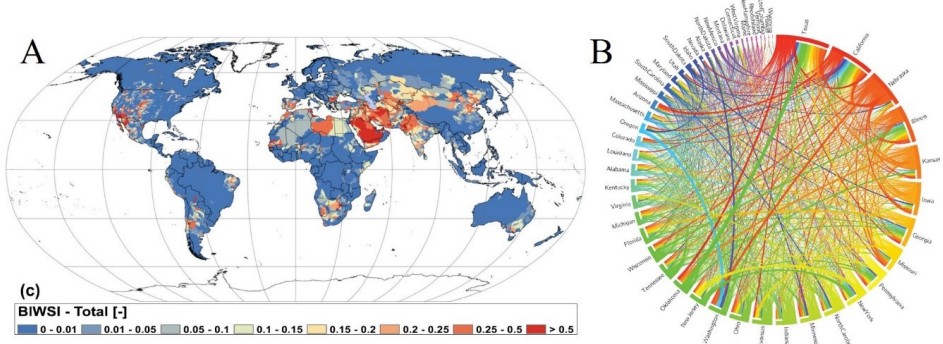

**Figure 3. Water footprint studies.** Water footprint studies exploit hydrological models to estimate blue and green water use in agriculture. This data is combined with data on food trade to estimate the fluxes of virtual water embedded in food trade. **A)** The unsustainable water footprint of agriculture is shown in red, where groundwater abstraction exceeds aquifer recharge (*taken from* Wada and Bierkens, 2014). **B)** Virtual water flows within the United States. U.S. States are ranked according to the total trade volume and plotted clockwise in descending order. The size of the outer bar indicates the total virtual water trade volume of each State as a percentage of total U.S. trade. Destination volume is indicated with links emanating from the outer bar of the same colour. Origin volume is indicated with a white area separating the outer bar from links of a different colour (*taken from* Dang et al., 2015).

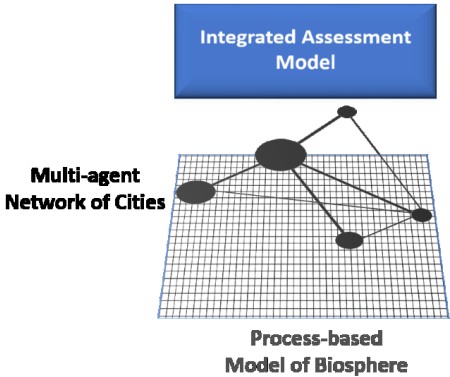

**Figure 4. Modelling Framework.** Our framework consists of three main components: An Integrated Assessment Model and a model of the biophysical environment which are coupled via a multi-agent network of cities and trade links. Agents in the multi-agent network receive environmental conditions from the biosphere model and the socioeconomic constraints to exploit those environmental constraints from the IAM. Agents trade resources via infrastructural trade network of roads, rail and shipping routes.



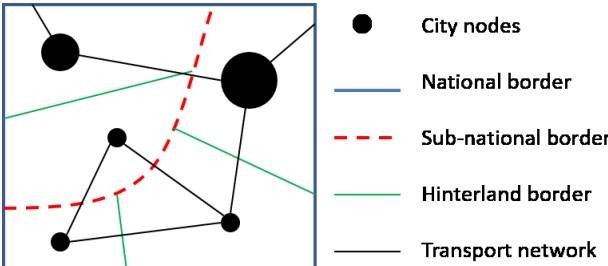

**Figure 5. City agent properties.** City agent hinterlands are defined by a hierarchical overlay of supra-sub-national boundaries and Theissen interpolation among city nodes based on cost distance via road, rail and ship. Cities are linked by physical infrastructural network of roads, rail and shipping routes.

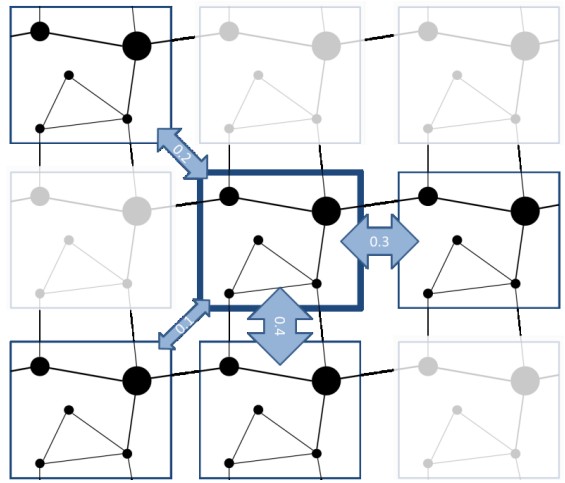

**Figure 6. Virtual Water Trade Network.** The trade network is hierarchical, containing an upper-level network that determines the strength of trade links among countries and a lower level physical infrastructural network which captures the network of roads, rail and shipping lanes that link cities. The link weights of the upper level network are calculated using a CGE model based on factors such as historical trade patterns, trade agreements etc. All virtual water flows via the physical trade network. The probability of trade among cities in the same country is based on supply, demand and cost-distance. The probability of trade among cities in different countries is based on supply, demand, cost-distance and link weights from the upper level network of bilateral trade patterns.





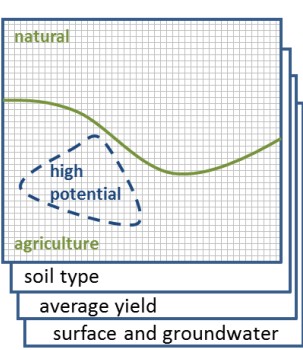

**Figure 7. Food production decisions.** City agents decide on agricultural expansion (left side) or intensification (right side) based on demand, spatially explicit crop production potential in their hinterlands and agroeconomic constraints.

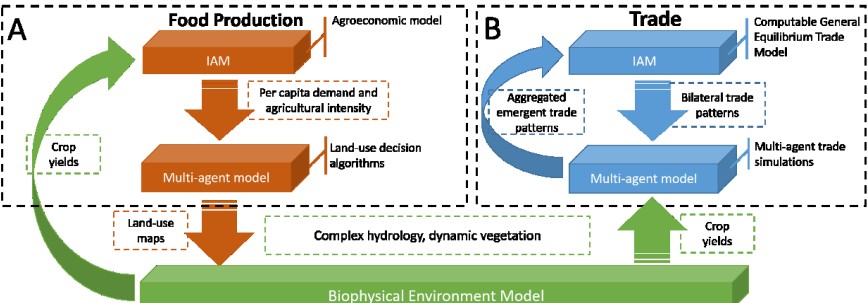

**Figure 8. Modelling Framework Workflow.** The multi-agent network sits at the interface between the Integrated Assessment Model (IAM) and a model of the biophysical environment **(A)** The IAM, calculates per capita food demand at a regional scale, which is converted into spatially explicit demand using population maps. The IAM also calculates agricultural intensity constraints. Based on agroeconomic and environmental constraints, agents make food production decisions which change the biophysical environment. The emergent crop production patterns are aggregated and feedback to the IAM **(B)** The IAM calculates bilateral trade patterns using Computable General Equilibrium trade model which constrains trade among cities in different countries and regions. Virtual Water Trade among cities is calculated based on supply and demand and cost-distance of trade via the physical infrastructural network. The emergent aggregated trade patterns feedback to CGE as input for the next simulation year.