# Peer review of "A framework for modelling the complexities of food and water security under globalisation"

_Earth System Dynamics, 2017_

## Referee Comment (RC1) · Anonymous Referee #1 · 9 May 2017

This is an interesting and timely paper, proposing a new method/framework to capture cross-scale and cross-sector, globalized food-water interactions as apparent, for example, in virtual water trade. It is great to see such a forward-looking paper that promotes novel ways of modelling (building on combinations of existing approaches). However, the text is rather long and it is difficult to comprehend what the core and the novel aspects of the new framework are about. Thus I recommend rewriting some sections, i.e. coming up with a better and more concise paper structure that much earlier introduces the main aspects of the framework, also introducing cities and their hinterlands as a focus/example. Below are some comments on where and how such a better focus could be arranged.

[Figure]

Abstract: "The approach unifies and extends the existing fields of hydrology, Integrated Assessment Modelling and agent-based modelling." This may be an overstatement... maybe not unifying but combining, integrating certain aspects of, or something like this. I think the present concept is not yet as mature.

Why have this first paragraph of the Introduction (which is more on water stress than the questions addressed here).

p 3 l 15: really unifies existing model approaches? see comment above

p 5 l 17: twice "it challenging"

p 8 l 34: "envisage a collection of basin-scale sociohydrology's", what do you mean?

Section 2 reads like a review of literature (2.1, cities in global context; 2.2, feedbacks of projections in general; 2.3 food and trade; 2.4 water use pathways – all three sub-sections only loosely connected by the way). It would be good if the overall idea of your (new) concept was summarized earlier and more systematically, so that readers know the particular context of this section. Also, in section 2.1, cities appear rather suddenly as a topic, please introduce this focus earlier.

Similarly section 3: lot of literature review (also including process descriptions that would better fit section 2) while one rather assumes that this section better guides the reader how and for what purpose earlier modelling/accounting approaches (i.e. footprint/virtual water trade studies and sociohydrological studies, ABM-based studies) could be unified. I recommend that these two sections be shortened, more focused, as they are quite verbose.

Section 4 also lacks some introductory remark on how all the aspects (or which of them) mentioned before find their way in a unified model framework. The claimed purpose that it will "capture the structure and constraints of the food system and the dynamics that operate within these constraints and bring about emergent water use patterns" is rather general and probably too ambitious (really capturing all the structure

and constraints of the food system? this would include many more aspects than those mentioned, including e.g. access). The basic structure of the framework needs to be clarified much earlier, otherwise it is difficult to follow what it actually covers, and how it does so. Section 4.2 starts with introducing yet other model types (water & food models), so this should rather go to section 3. Then follow again some process descriptions (cities linked to hinterlands) which should rather go to section 2?

And: is any of this new model and analysis framework already in operation, or is it 'just' a concept not yet tested?

p 7 l 26-27: is that really substantiated, "much of the mid-latitudes will become much less agriculturally productive"?

p 10 l 35: Basel, hinterland crossing three nations: I am not surprised by this and would expect that it actually extends across many more countries (because as you say earlier, industrialized countries import most of their products – from many countries around the world)?

p 11 l 12: "Thomas Brinkhoff": there are more such cases where citation is not correct (only surname to be used).

---

## Referee Comment (RC2) · Anonymous Referee #2 · 17 May 2017

This is a timely paper presenting a framework for modelling the water-food nexus in a globalised world. The topic goes across disciplines and is relevant for the ESD journal.

GENERAL COMMENT

While the approach proposed here is scientifically interesting, the structure of the paper does not seem appropriate. Also, some statements are far too strong, e.g. "the approach unifies and extends the existing fields of hydrology, Integrated Assessment Modelling and agent-based modelling". My general suggestion is to focus on the framework and its novel aspects, without making overstatements about it. I report below some specific comments that I hope can help improve the description of the proposed framework.

**SPECIFIC COMMENTS**

1) The modelling framework, which is the core of this paper, comes abruptly after 10 pages of literature review. I propose to introduce it early on, provide more details about the framework (e.g. have you tried to build an actual model based on this?), while avoiding too much text for literature review (Chapter 2 and 3 are really too long).

2) Introduction: Is the text up to line 34 really needed? It is very basic, it reads like a textbook and it is not much related with the framework.

3) An entire section of sociohydrology (Section 3.2) seems a bit too much here, as the proposed framework is in fact an upgrade of IAM coupled with a biophysical model. In any case, while I agree that "sociohydrological studies so far have explained the observed emergent phenomena by allowing human agency...", there are studies of this kind that were published before 2014. So, if reference to sociohydrology is really needed, previous efforts made by other scholars should not be ignored here. Also, if there is an entire section in Chapter 3 about sociohydrology, there should be at least an entire paragraph later on in the paper discussing the link between the proposed framework and sociohydrological research.

4) The paper states that "currently 30% of energy produced is used in food production, with fluctuations in energy costs having direct impacts on agriculture and thus water resources". Yet, the interlink with energy production is then almost forgotten in the rest of the paper. I understand the focus on food, but the water-food-energy nexus cannot be completely neglected.

5) What do the authors exactly mean by resilience/resilient and sustainability/sustainable? These "buzzwords" are used a number of times, but in different contexts and, in my opinion, with a completely different meaning. I'm fine with any definition, as long as these terms are used consistently throughout the entire article. Still, I must confess that I feel a bit uncomfortable to figure out the exact meaning of statements like "the optimally resilient and sustainable solution".
* * *
6) Typos: "it challenging it challenging", references with first name abbreviation, etc...

---

## Referee Comment (RC3) · Anonymous Referee #3 · 26 May 2017

General comments

Although Integrated Assessment Models (IAMs) are powerful tools to investigate complex long-term issues of global change, their coarse spatial resolution hampers effective treatment of spatiotemporally heterogenous phenomena such as water constraint in food production. The framework proposed in this manuscript has a potential capability to relieve this limitation by expressing the cities as agents that are interlinked with transportation network and receiving information from both IAM and spatially detailed biophysical models.

If a numerical model was successfully developed based on this framework, it would largely enhance the capability of IAMs. Consequently, it would contribute to seek prac-

tical solutions for complex global issues such as achievement of the Paris Agreement and the Sustainable Development Goals, which will be an important advancement in the global environmental science. Nonetheless, it was hard for me to comment on this manuscript as a referee because this paper only shows the framework of a forthcoming model (i.e. I took this paper as an elaborated research plan). One can hardly judge the validity of the authors' framework unless the concept is actually implemented and validated. What I could do was to comment on the validity of logical flow of the paper.

I observe two major concerns in the logic of this paper. First, the authors little refer to the published land use models. Land use models allocate land use under given socio-economic conditions and shocks which largely overlap with the key concepts and functions of the authors' framework. For example, Lotze-Campen et al. (2008), Wise et al. (2009), Konar et al. (2013), Hejazi et al. (2015), Bonsch et al. (2016), and Hasegawa et al. (2017) have already resolved multiple challenges raised by the authors. It should be more clearly elaborated what are the literary unresolved challenges of IAMs and what would be the key differences between the approaches of the forerunners and the authors. Further focused review should be added to text. Second, I am wondering the authors may overvalue the international food trade. Although important, for example, the fraction of the traded major grains to the total production is approximately 15% in 2005. A major part of food production is consumed domestically. An excessive emphasis on trade might distort the reality. Further discussion should be added on non-traded food production and water use.

Specific comments

Page 2 Line 14 "The redistribution of food via trade is central to determining water resources use": I don't believe this statement is right. Only a limited portion of food is internationally traded, and it only partly determines the water. I would like to see here the total production of agricultural products and the fraction of internationally traded. I believe similar figures can be easily made using the total water use for food production (NB: include green water as well) by consulting earlier works (e.g. Aldaya et al., 2010;

Hanasaki et al. 2010; Hoff et al. 2010; Fader et al. 2011; Gerten et al. 2011).

Page 7 Line 18 "Water footprint studies . . . (van Beek et al. 2011; Wada et al. 2011)": I don't believe these two papers are on water footprint. The works by Hanasaki et al. (2010) and Fader et al. (2011) are more directly relevant in this context. Hanasaki (2016) provides an overview of the water footprint studies by applying global hydrological models.

Page 8 "3.2 Sociohydrological Studies": I hardly found any direct or concrete linkage of the sociohydrology and the framework proposed in this study. I would see more focused discussion why and how sociohydrogy is relevant to this study.

Page 12 "Food production and water use": As mentioned in General Comments, a review on earlier efforts linking water-land-food models and IAMs seems largely missing here. I note that earlier studies seldom applied agent-based model (ABM), but still clarifications are needed what has been achieved without ABM, and what would be potentially achieved by adopting ABM based on a fair literature survey.

Page 14 "The framework can be applied across scales to investigate changes at catchment, city or global scale": The statement sounds a bit too strong since no concrete evidence of the capability of framework is presented in this paper. The dominant force or process of linkage between cities would be substantially different across scales. For instance, even if the connection between New York and London and that of Seoul and its commuter towns can be both expressed as nodes and links, their link must be formulated fundamentally differently. More specifically, local connections are strongly influenced by local non-market circumstances such as regulations, custom, and cultures, which is hardly obtained from neither IAMs nor biophysical models. If you wish to keep this argument, elaborate how the scale issues would be basically resolved.

References

Aldaya, M. M., Allan, J. A., and Hoekstra, A. Y.:   Strategic importance of

green water in international crop trade, Ecological Economics, 69, 887-894, http://dx.doi.org/10.1016/j.ecolecon.2009.11.001, 2010.

Bonsch, M., Humpenöder, F., Popp, A., Bodirsky, B., Dietrich, J. P., Rolinski, S., Biewald, A., Lotze-Campen, H., Weindl, I., Gerten, D., and Stevanovic, M.: Trade-offs between land and water requirements for large-scale bioenergy production, GCB Bioenergy, 8, 11-24, 10.1111/gcbb.12226, 2016.

Fader, M., Gerten, D., Thammer, M., Heinke, J., Lotze-Campen, H., Lucht, W., and Cramer, W.: Internal and external green-blue agricultural water footprints of nations, and related water and land savings through trade, Hydrol. Earth Syst. Sci., 15, 1641-1660, 10.5194/hess-15-1641-2011, 2011.

Gerten, D., Heinke, J., Hoff, H., Biemans, H., Fader, M., and Waha, K.: Global water availability and requirements for future food production, J. Hydromet., 12, 885-899, 10.1175/2011JHM1328.1, 2011.

Hanasaki, N., Inuzuka, T., Kanae, S., and Oki, T.: An estimation of global virtual water flow and sources of water withdrawal for major crops and livestock products using a global hydrological model, J. Hydrol., 384, 232-244, 10.1016/j.jhydrol.2009.09.028, 2010.

Hanasaki, N.: Estimating Virtual Water Contents Using a Global Hydrological Model: Basis and applications, in: Terrestrial Water Cycle and Climate Change: Natural and Human-Induced Impacts, edited by: Tang, Q., and Oki, T., John Wiley & Sons, Inc., 209-228, 2016.

Hasegawa, T., Fujimori, S., Ito, A., Takahashi, K., and Masui, T.: Global land-use allocation model linked to an integrated assessment model, Science of The Total Environment, 580, 787-796, https://doi.org/10.1016/j.scitotenv.2016.12.025, 2017.

Hejazi, M. I., Voisin, N., Liu, L., Bramer, L. M., Fortin, D. C., Hathaway, J. E., Huang, M., Kyle, P., Leung, L. R., Li, H.-Y., Liu, Y., Patel, P. L., Pulsipher, T. C., Rice, J. S., Tesfa, T.

K., Vernon, C. R., and Zhou, Y.: 21st century United States emissions mitigation could increase water stress more than the climate change it is mitigating, P. Natl. Acad. Sci. USA, 10.1073/pnas.1421675112, 2015.

Hoff, H., Falkenmark, M., Gerten, D., Gordon, L., Karlberg, L., and Rockström, J.: Greening the global water system, J. Hydrol., 384, 177-186, http://dx.doi.org/10.1016/j.jhydrol.2009.06.026, 2010.

Konar, M., Hussein, Z., Hanasaki, N., Mauzerall, D. L., and Rodriguez-Iturbe, I.: Virtual water trade flows and savings under climate change, Hydrol. Earth Syst. Sci., 17, 3219-3234, 10.5194/hess-17-3219-2013, 2013.

Lotze-Campen, H., Müller, C., Bondeau, A., Rost, S., Popp, A., and Lucht, W.: Global food demand, productivity growth, and the scarcity of land and water resources: a spatially explicit mathematical programming approach, Agr. Econ., 39, 325-338, 10.1111/j.1574-0862.2008.00336.x, 2008.

Wise, M., Calvin, K., Thomson, A., Clarke, L., Bond-Lamberty, B., Sands, R., Smith, S. J., Janetos, A., and Edmonds, J.: Implications of Limiting $CO_2$ Concentrations for Land Use and Energy, Science, 324, 1183-1186, 10.1126/science.1168475, 2009.

---

## Author Comment (AC1) · 4 Jul 2017

**Response to Reviewer 1**

We thank anonymous reviewer 1 for their considered and constructive comments on our manuscript "A framework for modelling the complexities of food and water security under globalisation". Following is our response. The reviewer's comments are written in *Italics*.

*This is an interesting and timely paper, proposing a new method/framework to capture cross-scale and cross-sector, globalized food-water interactions as apparent, for example, in virtual water trade. It is great to see such a forward-looking paper that promotes novel ways of modelling (building on combinations of existing approaches). However, the text is rather long and it is difficult to comprehend what the core and the novel aspects of the new framework are about. Thus, I recommend rewriting some sections, i.e. coming up with a better and more concise paper structure that much earlier introduces the main aspects of the framework, also introducing cities and their hinterlands as a focus/ example. Below are some comments on where and how such a better focus could be arranged.*

We agree in reflection that the paper can benefit from a more concise structure which presents the core elements of the framework earlier. We have rewritten section 1 to immediately outline the knowledge gaps in understanding food and water security under globalisation that we set out to fill with our framework. Namely, it remains a knowledge gap to capture regional and sectoral interdependencies and cross-scale feedbacks associated with food and water security within a single model framework. Following this, we introduce the main aspects of the framework: cities and hinterlands and the networks that connect them. We explain why these are important elements to capture in order to understand food and water security under globalisation.

*Abstract: "The approach unifies and extends the existing fields of hydrology, Integrated Assessment Modelling and agent-based modelling." This may be an overstatement: maybe not unifying but combining, integrating certain aspects of, or something like this. I think the present concept is not yet as mature.*

In the revised manuscript, this is rewritten as "The framework integrates aspects of existing models and approaches in the fields of hydrology, Integrated Assessment Modelling and agent-based modelling"

*Why have this first paragraph of the Introduction (which is more on water stress than the questions addressed here).*

We agree and have removed the first 2 paragraphs of the introduction from the revised manuscript and replaced with an introductory paragraph that outlines the knowledge gaps within food and water security under globalisation that our framework sets out to address.

*p 3 l 15: really unifies existing model approaches? see comment above*

We have removed this from the revised manuscript

*p 5 l 17: twice "it challenging"*

Corrected

*p 8 l 34: "envisage a collection of basin-scale sociohydrology's", what do you mean?*

We have made the section on sociohydrology more concise, as a result the phrase quoted here has been omitted in the revised manuscript.

*Section 2 reads like a review of literature (2.1, cities in global context; 2.2, feedbacks of projections in general; 2.3 food and trade; 2.4 water use pathways – all three subsections only loosely connected by the way). It would be good if the overall idea of your (new) concept was summarized earlier and more systematically, so that readers know the particular context of this section. Also, in section 2.1, cities appear rather suddenly as a topic, please introduce this focus earlier.*

In the revised manuscript, we have integrated section 2 and 3 into a new section 2 which is more concise. In the new section 2, we have focused on three core topics which we feel are key to understanding water resource use within the globalised food system. These are regional interdependence, sectoral interdependence and cross-scale feedbacks. We outline how these are addressed to differing extents in existing models and approaches and the knowledge gaps in those approaches that we set out to close with our framework. Namely, integrating regional and sectoral interdependencies and cross-scale feedbacks within a single model framework.  In the new section 3 in which we present our framework in detail, we systematically outline how our framework can fill the knowledge gaps outlined in section 2.

*Similarly section 3: lot of literature review (also including process descriptions that would better fit section 2) while one rather assumes that this section better guides the reader how and for what purpose earlier modelling/accounting approaches (i.e. footprint/virtual water trade studies and sociohydrological studies, ABM-based studies) could be unified. I recommend that these two sections be shortened, more focused, as they are quite verbose.*

See previous remark

*Section 4 also lacks some introductory remark on how all the aspects (or which of them) mentioned before find their way in a unified model framework. The claimed purpose that it will "capture the structure and constraints of the food system and the dynamics that operate within these constraints and bring about emergent water use patterns" is rather general and probably too ambitious (really capturing all the structure and constraints of the food system? this would include many more aspects than those mentioned, including e.g. access). The basic structure of the framework needs to be clarified much earlier, otherwise it is difficult to follow what it actually covers, and how it does so. Section 4.2 starts with introducing yet other model types (water & food models), so this should rather go to section 3. Then follow again some process descriptions (cities linked to hinterlands) which should rather go to section 2?*

In the revised manuscript, all information about other models and processes has been moved to section 2 which covers knowledge gaps in current approaches. We have expanded on the description of land use models (reviewer 3) and placed this in the section 2 of the revised manuscript.

In the revised manuscript, the framework is introduced for the first time in the 2nd paragraph of the introduction section 1. Section 3 of the revised manuscript presents a detailed description the framework. We have removed general terms such as "capture the structure and constraints of the food system and the dynamics that operate within these constraints and bring about emergent water use patterns". Instead, we provide detailed and specific descriptions of the aspects of the global food system the framework sets out to capture.

*And: is any of this new model and analysis framework already in operation, or is it 'just' a concept not yet tested?*

Elements are in operation already. In the revised manuscript, we are explicit about the level of development of each component of the framework. This will is shown in a revised version of figure 8.

*p 7 l 26-27: is that really substantiated, "much of the mid-latitudes will become much less agriculturally productive"?*

In the revised manuscript, we have moderated the tone of the statement as follows: "studies indicate that unsustainable groundwater abstraction in mid-latitude regions threatens future food security". This is substantiated by studies that show that an increasing proportion of irrigated agriculture in mid-latitude countries is sustained by unsustainable groundwater abstraction (see Dalin et al., 2017; Wada et al., 2012).

*p 10 l 35: Basel, hinterland crossing three nations: I am not surprised by this and would expect that it actually extends across many more countries (because as you say earlier, industrialized countries import most of their products – from many countries around the world)?*

We have removed the specific example of Basel from the revised manuscript. As we mention in the discussion manuscript (P10, L34), if policy between two hinterlands stimulates free trade, then the effective hinterlands of those cities may expand.

*p 11 l 12: "Thomas Brinkhoff": there are more such cases where citation is not correct (only surname to be used).*

Examples such as this have been corrected in the revised manuscript

**References**

Dalin, C., Wada, Y., Kastner, T., Puma, M.J., 2017. Groundwater depletion embedded in international food trade. Nature 543, 700–704. doi:10.1038/nature21403

Wada, Y., van Beek, L.P.H., Bierkens, M.F.P., 2012. Nonsustainable groundwater sustaining irrigation: A global assessment. Water Resour. Res. 48, W00L06. doi:10.1029/2011WR010562

---

## Author Comment (AC2) · 4 Jul 2017

**Response to Reviewer 2**

We thank anonymous reviewer 2 for their considered and constructive comments on our manuscript "A framework for modelling the complexities of food and water security under globalisation". Following is our response. The reviewer's comments are written in *Italics*.

*This is a timely paper presenting a framework for modelling the water-food nexus in a globalised world. The topic goes across disciplines and is relevant for the ESD journal.*

*GENERAL COMMENT*

*While the approach proposed here is scientifically interesting, the structure of the paper does not seem appropriate. Also, some statements are far too strong, e.g. "the approach unifies and extends the existing fields of hydrology, Integrated Assessment Modelling and agent-based modelling". My general suggestion is to focus on the framework and its novel aspects, without making overstatements about it. I report below some specific comments that I hope can help improve the description of the proposed framework.*

*SPECIFIC COMMENTS*

*1) The modelling framework, which is the core of this paper, comes abruptly after 10 pages of literature review. I propose to introduce it early on, provide more details about the framework (e.g. have you tried to build an actual model based on this?), while avoiding too much text for literature review (Chapter 2 and 3 are really too long).*

We agree in reflection that the paper can benefit from a more concise structure which presents the core elements of the framework earlier. In the revised manuscript, we have rewritten section 1 to immediately outline the knowledge gaps in understanding food and water security under globalisation that we set out to fill with our framework. Namely, it remains a knowledge gap to capture regional and sectoral interdependencies and cross-scale feedbacks associated with food and water security within a single model framework. Following this, we introduce the main aspects of the framework: cities and hinterlands and the networks that connect them. We explain why these are important elements to capture in order to understand food and water security under globalisation.

In addition, in the revised manuscript we have integrated section 2 and 3 into a new section 2 which is more concise. In the new section 2, we have focused on explaining how regional interdependence, sectoral interdependence and cross-scale feedbacks are captured to differing extents in existing models and approaches. We then outline the knowledge gaps in existing approaches that our framework sets out to fill. Namely, integrating regional and sectoral interdependencies and cross-scale feedbacks within a single model framework.

The model is not yet built. However, elements are in operation already. In the revised manuscript, we are explicit about the level of development of each component of the framework. This will be shown in a revised version of figure 8.

*2) Introduction: Is the text up to line 34 really needed? It is very basic, it reads like a textbook and it is not much related with the framework.*

We agree with reviewer 2 and have deleted this text in the revised manuscript.

*3) An entire section of sociohydrology (Section 3.2) seems a bit too much here, as the proposed framework is in fact an upgrade of IAM coupled with a biophysical model. In any case, while I agree that "sociohydrological studies so far have explained the observed emergent phenomena by allowing*

*human agency: : :", there are studies of this kind that were published before 2014. So, if reference to sociohydrology is really needed, previous efforts made by other scholars should not be ignored here. Also, if there is an entire section in Chapter 3 about sociohydrology, there should be at least an entire paragraph later on in the paper discussing the link between the proposed framework and sociohydrological research.*

In the revised manuscript, we have integrated section 2 and 3 into a new section 2 which is more concise. The text given over sociohydrology is shortened and included in the subsection dealing with cross-scale feedbacks (new section 2.3). In this section, we outline that sociohydrology studies set out to understand cross-scale spatiotemporal feedbacks by capturing how short term or small-scale interactions between humans and the environment can bring about long term and large scale emergent changes in water resources (Sivapalan et al., 2012; Sivapalan and Blöschl, 2015). We highlight that sociohydrological studies suffer from a disciplinary focus on water and do not capture important sectoral interdependencies. Equally, they have so far assumed the systems of concern are isolated entities in space, e.g., an agricultural river basin whereas, in a globalised world, many different such entities may interdependent with other regions owing to trade in goods (e.g., food). In the discussion section of the revised manuscript, we add a short discussion about how sociohydrological studies can benefit from incorporating regional and sectoral interdependence in order to better understand human-water dynamics in a globalised world.

In the revised manuscript, have made efforts to include a more comprehensive literature review, incorporating literature prior to 2014.

*4) The paper states that "currently 30% of energy produced is used in food production, with fluctuations in energy costs having direct impacts on agriculture and thus water resources". Yet, the interlink with energy production is then almost forgotten in the rest of the paper. I understand the focus on food, but the water-food-energy nexus cannot be completely neglected.*

In the revised manuscript, we have focused on three core topics: regional interdependence, sectoral interdependence and cross-scale feedbacks. The water-food-energy nexus falls under sectoral interdependence and we have expanded our discussion of this important topic in section 2 of the revised manuscript.

*5) What do the authors exactly mean by resilience/resilient and sustainability/ sustainable? These "buzzwords" are used a number of times, but in different contexts and, in my opinion, with a completely different meaning. I'm fine with any definition, as long as these terms are used consistently throughout the entire article. Still, I must confess that I feel a bit uncomfortable to figure out the exact meaning of statements like "the optimally resilient and sustainable solution".*

We have provided definitions for these terms in the revised manuscript so readers are clear on what statements like "optimally resilient and sustainable solution" mean.

*6) Typos: "it challenging it challenging", references with first name abbreviation, etc...#*

Corrected in the revised manuscript

**References**

Sivapalan, M., Blöschl, G., 2015. Time scale interactions and the coevolution of humans and water. Water Resour. Res. 51, 6988–7022. doi:10.1002/2015WR017896
Sivapalan, M., Savenije, H.H.G., Blöschl, G., 2012. Socio-hydrology: A new science of people and water. Hydrol. Process. 26, 1270–1276. doi:10.1002/hyp.8426

---

## Author Comment (AC3) · 4 Jul 2017

**Response to Reviewer 3**

We thank anonymous reviewer 3 for their considered and constructive comments on our manuscript "A framework for modelling the complexities of food and water security under globalisation". Following is our response. The reviewer's comments are written in *Italics*.

*General comments*

*Although Integrated Assessment Models (IAMs) are powerful tools to investigate complex long-term issues of global change, their coarse spatial resolution hampers effective treatment of spatiotemporally heterogenous phenomena such as water constraint in food production. The framework proposed in this manuscript has a potential capability to relieve this limitation by expressing the cities as agents that are interlinked with transportation network and receiving information from both IAM and spatially detailed biophysical models.*

*If a numerical model was successfully developed based on this framework, it would largely enhance the capability of IAMs. Consequently, it would contribute to seek practical solutions for complex global issues such as achievement of the Paris Agreement and the Sustainable Development Goals, which will be an important advancement in the global environmental science. Nonetheless, it was hard for me to comment on this manuscript as a referee because this paper only shows the framework of a forthcoming model (i.e. I took this paper as an elaborated research plan). One can hardly judge the validity of the authors' framework unless the concept is actually implemented and validated. What I could do was to comment on the validity of logical flow of the paper.*

We are glad that the reviewer has provided comments on the paper despite the model not yet being operational. We believe that exposing our vision to critical peer review and a wider audience will help develop the ideas put forward, further. It is also our hope that the framework outlined here will stimulate debate, innovation and new ideas about how to move the discipline of global change modelling forward.

We begin by stating that in the revised manuscript, we have rewritten section 1 to immediately outline the knowledge gaps in understanding food and water security under globalisation that we set out to fill with our framework. Namely, it remains a knowledge gap to capture regional and sectoral interdependencies and cross-scale feedbacks associated with food and water security within a single model framework. Following this, we introduce the main aspects of the framework: cities and hinterlands and the networks that connect them. We explain why these are important elements to capture in order to understand food and water security under globalisation.

In addition, in the revised manuscript we have integrated section 2 and 3 into a new section 2 which is more concise. In the new section 2, we have focused on explaining how regional interdependence, sectoral interdependence and cross-scale feedbacks are captured to differing extents in existing models and approaches. We then outline the knowledge gaps in existing approaches that our framework sets out to fill. Namely, integrating regional and sectoral interdependencies and cross-scale feedbacks within a single model framework.

*I observe two major concerns in the logic of this paper. First, the authors little refer to the published land use models. Land use models allocate land use under given socio-economic conditions and shocks which largely overlap with the key concepts and functions of the authors' framework. For example, Lotze-Campen et al. (2008), Wise et al. (2009), Konar et al. (2013), Hejazi et al. (2015), Bonsch et al. (2016), and Hasegawa et al. (2017) have already resolved multiple challenges raised by the authors. It should be more clearly elaborated what are the literary unresolved challenges of IAMs and what would*

*be the key differences between the approaches of the forerunners and the authors. Further focused review should be added to text.*

In section 2 of the revised manuscript, we have provided a comprehensive review of land use model approaches. We outline the progress made in understanding land use change from these approaches and specific challenges that remain, which can be addressed to a certain extent with our approach. Specifically, we highlight that although existing approaches capture regional and sectoral interdependencies or cross-scale feedbacks to differing extents, capturing these factors within one framework has yet to be accomplished, to our knowledge. We outline how our framework can provide a means to begin to close this knowledge gap.

*Second, I am wondering the authors may overvalue the international food trade. Although important, for example, the fraction of the traded major grains to the total production is approximately 15% in 2005. A major part of food production is consumed domestically. An excessive emphasis on trade might distort the reality. Further discussion should be added on non-traded food production and water use.*

By trade we mean international and domestic trade. In the revised manuscript, we provide a definition of what we mean by trade to avoid confusion. As we mention in the discussion manuscript (Page 2, line 28-31), the magnitude of virtual water / food trade is underestimated because most studies focus only on international trade. In section 2 of the revised manuscript, we underline that the domestic trade fluxes are an important knowledge gap in existing approaches that our framework sets out to close. Our framework sets out to begin to close this knowledge gap by capturing spatially explicit demand and production potential as well as infrastructural networks which constrain the how food is redistributed within countries to meet demand and the associated impact on food and water security.

We have included discussion of non-traded food in section 2 of the revised manuscript.

*Specific comments*

*Page 2 Line 14 "The redistribution of food via trade is central to determining water resources use": I don't believe this statement is right. Only a limited portion of food is internationally traded, and it only partly determines the water. I would like to see here the total production of agricultural products and the fraction of internationally traded. I believe similar figures can be easily made using the total water use for food production (NB: include green water as well) by consulting earlier works (e.g. Aldaya et al., 2010; Hanasaki et al. 2010; Hoff et al. 2010; Fader et al. 2011; Gerten et al. 2011).*

We underline that 54% of people currently live in urban areas and depend on domestic or international trade for food security. Thus, we argue that domestic and international trade plays an important role in determining water resource use (see chapter 6 of the United Nations Water Development Report 2015: Water for a Sustainable World).

In the revised manuscript, we provide examples of the amount of green and blue water used in food production and embedded in traded food based on the references recommended by reviewer 3.

*Page 7 Line 18 "Water footprint studies : : : (van Beek et al. 2011; Wada et al. 2011)": I don't believe these two papers are on water footprint. The works by Hanasaki et al. (2010) and Fader et al. (2011) are more directly relevant in this context. Hanasaki (2016) provides an overview of the water footprint studies by applying global hydrological models.*

We have included the recommended literature in the revised manuscript.

*Page 8 "3.2 Sociohydrological Studies": I hardly found any direct or concrete linkage of the sociohydrology and the framework proposed in this study. I would see more focused discussion why and how sociohydrogy is relevant to this study.*

In the revised manuscript, we have focused on three core topics which we feel are key to understanding water resource use within the globalised food system. These are regional interdependence, sectoral interdependence and cross-scale feedbacks. In section 2 we discuss how existing models and approaches capture each of these to differing extents and highlight knowledge gaps in existing approaches that our framework sets out to fill. The text given over to sociohydrology is shortened and included in subsection 2.3 dealing with cross-scale feedbacks. In this section, we outline that sociohydrology studies set out to understand cross-scale spatiotemporal feedbacks by capturing how short term or small-scale interactions between humans and the environment can bring about long term and large scale emergent changes in water resources (Sivapalan et al., 2012; Sivapalan and Blöschl, 2015). We highlight that sociohydrological studies suffer from a disciplinary focus on water and do not capture important sectoral interdependencies. Equally, they have so far assumed the systems of concern are isolated entities in space, e.g., an agricultural river basin, whereas in a globalised world, many different such entities may interdependent with other regions owing to trade in goods (e.g., food). In the discussion section of the revised manuscript, we add a short discussion about how sociohydrological studies can benefit from incorporating regional and sectoral interdependence in order to better understand human-water dynamics in a globalised world.

*Page 12 "Food production and water use": As mentioned in General Comments, a review on earlier efforts linking water-land-food models and IAMs seems largely missing here. I note that earlier studies seldom applied agent-based model (ABM), but still clarifications are needed what has been achieved without ABM, and what would be potentially achieved by adopting ABM based on a fair literature survey.*

In section 2 of the revised manuscript, we have expanded the literature review on work done to link water, land and food within IAMs. We outline specific gaps in knowledge in these studies that our framework sets out to fill.

*Page 14 "The framework can be applied across scales to investigate changes at catchment, city or global scale": The statement sounds a bit too strong since no concrete evidence of the capability of framework is presented in this paper. The dominant force or process of linkage between cities would be substantially different across scales. For instance, even if the connection between New York and London and that of Seoul and its commuter towns can be both expressed as nodes and links, their link must be formulated fundamentally differently. More specifically, local connections are strongly influenced by local non-market circumstances such as regulations, custom, and cultures, which is hardly obtained from neither IAMs nor biophysical models. If you wish to keep this argument, elaborate how the scale issues would be basically resolved.*

We agree with the reviewer that the above statement is too strong and have removed it from the revised paper. Nonetheless, we elaborate on the reviewer's comments below. We have strived to clarify these issues in the explanation of the framework in section 3 of the revised manuscript.

In terms of trade regulation, as stated in the discussion manuscript (P10, Line 25-35), we define city hinterlands based on the hierarchical overlay of supra-sub national administrative borders and theissen polygon operation among cities based on cost-distance of trade via road, rail and inland water ways. Our framework thus provides a structure that can capture regulation at the scale of an administrative region, where the data are available. Currently CGEs used in IAMs contain regulatory

data at the scale of countries or regions. If regulatory data is available at a finer scale, our framework provides a structure to incorporate that data. If regulations between two hinterlands stimulate free trade, then the effective hinterlands of those cities may expand. We have provided a more detailed version of figure 5 in the revised manuscript to illustrate the how a city and its hinterland are defined.

In terms of issues such as customs and culture, the framework described does not provide a structure to capture those societal elements at this time.

Following, we provide an illustration of how an executed version of the model framework would simulate trade between New York and Tokyo as opposed to Seoul and its commuter cities. To take the example of New York and Tokyo first. The upper level network of socioeconomic trade links will constrain the probability of trade between Japan and the US based on the CGE. The share of that trade that will come from New York will depend on the production of goods in the hinterland of New York that meet Japanese demand. It will also depend on the competition among American cities and hinterlands to meet that demand. Given, the lower cost-distance for trade, a west coast city may be more likely to meet demand from Japan than a city on the east coast. If trade data becomes available at state level, the framework structure allows for that to be incorporated and a finer scale estimation of food and virtual water fluxes from the hinterland of New York city to Japan to be made.

Contrast that with Seoul and its commuter cities. Assuming the finest scale data on food production and consumption is at national-level, fluxes in food and virtual water among Korean cities will be estimated based solely on spatially explicit population demand, production potential and the cost-distance between cities based on transport cost along infrastructure networks. This provides an estimation of resource redistribution within Korea based on these factors. As finer-scale sub-national trade data become available, these redistribution estimates can be improved.

**References from reviewer**

*Aldaya, M. M., Allan, J. A., and Hoekstra, A. Y.: Strategic importance of green water in international crop trade, Ecological Economics, 69, 887-894, http://dx.doi.org/10.1016/j.ecolecon.2009.11.001, 2010.*

*Bonsch, M., Humpenöder, F., Popp, A., Bodirsky, B., Dietrich, J. P., Rolinski, S., Biewald, A., Lotze-Campen, H., Weindl, I., Gerten, D., and Stevanovic, M.: Tradeoffs between land and water requirements for large-scale bioenergy production, GCB Bioenergy, 8, 11-24, 10.1111/gcbb.12226, 2016.*

*Fader, M., Gerten, D., Thammer, M., Heinke, J., Lotze-Campen, H., Lucht, W., and Cramer, W.: Internal and external green-blue agricultural water footprints of nations, and related water and land savings through trade, Hydrol. Earth Syst. Sci., 15, 1641- 1660, 10.5194/hess-15-1641-2011, 2011.*

*Gerten, D., Heinke, J., Hoff, H., Biemans, H., Fader, M., and Waha, K.: Global water availability and requirements for future food production, J. Hydromet., 12, 885-899, 10.1175/2011JHM1328.1, 2011.*

*Hanasaki, N., Inuzuka, T., Kanae, S., and Oki, T.: An estimation of global virtual water flow and sources of water withdrawal for major crops and livestock products using a global hydrological model, J. Hydrol., 384, 232-244, 10.1016/j.jhydrol.2009.09.028, 2010.*

*Hanasaki, N.: Estimating Virtual Water Contents Using a Global Hydrological Model: Basis and applications, in: Terrestrial Water Cycle and Climate Change: Natural and Human-Induced Impacts, edited by: Tang, Q., and Oki, T., John Wiley & Sons, Inc., 209-228, 2016.*

*Hasegawa, T., Fujimori, S., Ito, A., Takahashi, K., and Masui, T.: Global land-use allocation model linked to an integrated assessment model, Science of The Total Environment, 580, 787-796, https://doi.org/10.1016/j.scitotenv.2016.12.025, 2017.*

*Hejazi, M. I., Voisin, N., Liu, L., Bramer, L. M., Fortin, D. C., Hathaway, J. E., Huang, M., Kyle, P., Leung, L. R., Li, H.-Y., Liu, Y., Patel, P. L., Pulsipher, T. C., Rice, J. S., Tesfa, T. K., Vernon, C. R., and Zhou, Y.: 21st century United States emissions mitigation could increase water stress more than the climate change it is mitigating, P. Natl. Acad. Sci. USA, 10.1073/pnas.1421675112, 2015.*

*Hoff, H., Falkenmark, M., Gerten, D., Gordon, L., Karlberg, L., and Rockström, J.: Greening the global water system, J. Hydrol., 384, 177-186, http://dx.doi.org/10.1016/j.jhydrol.2009.06.026, 2010.*

*Konar, M., Hussein, Z., Hanasaki, N., Mauzerall, D. L., and Rodriguez-Iturbe, I.: Virtual water trade flows and savings under climate change, Hydrol. Earth Syst. Sci., 17, 3219-3234, 10.5194/hess-17-3219-2013, 2013.*

*Lotze-Campen, H., Müller, C., Bondeau, A., Rost, S., Popp, A., and Lucht, W.: Global food demand, productivity growth, and the scarcity of land and water resources: a spatially explicit mathematical programming approach, Agr. Econ., 39, 325-338, 10.1111/j.1574-0862.2008.00336.x, 2008.*

*Wise, M., Calvin, K., Thomson, A., Clarke, L., Bond-Lamberty, B., Sands, R., Smith, S. J., Janetos, A., and Edmonds, J.: Implications of Limiting CO2 Concentrations for Land Use and Energy, Science, 324, 1183-1186, 10.1126/science.1168475, 2009.*

**References in rebuttal**

Sivapalan, M., Blöschl, G., 2015. Time scale interactions and the coevolution of humans and water. Water Resour. Res. 51, 6988–7022. doi:10.1002/2015WR017896

Sivapalan, M., Savenije, H.H.G., Blöschl, G., 2012. Socio-hydrology: A new science of people and water. Hydrol. Process. 26, 1270–1276. doi:10.1002/hyp.8426

The United Nations World Water Development Report 5: Water for a Sustainable World, 2015. Paris, UNESCO.

---

## Author Response (AR2)

Reply to Editor.

We very much thank the editor and 2 reviewers for the additional comments, which have helped improve the manuscript. We provide a response to the editor's comments below. Editors comments are in italics. Marked-up manuscript follows responses. Page and line numbers coincide with final manuscript without mark-up.

*Dear Authors*

*Thank you for your revised manuscript. I have decided to accept your manuscript subject to minor revisions that I will review.*

*Both reviewer reports are positive. Neither require corrections, but R2 provides a list of suggestions which I strongly recommend you consider. I have captured some of these and made additional recommendations below.*

*The Sustainable Development Goals may provide a useful framing for your methodology. How can we reduce hunger, provide clean water and reduce climate change impacts for example? This challenge seems to suggest an intergrated modelling approach. You briefly mention this at the end of the article, but it may serve as a more important framing near the start.*

We have done this, see Page 2, Line 3 of the revised manuscript

*P2 lin 25 Rockström et al., 2009, I suggest you use more recent Steffen et al (2015) as this also includes regional rather than global water use boundary.*

We have included this reference

*P3 L15. You use climate modelling as a metaphor for model complexity. But to avoid any possible confusion, I suggest you mention that socioeconomic processes have come to be linked to biophysical models that include climate dynamics: IAMs. More generally, you are arguing for the need to develop socio-ecological modelling. That is a reasonably well established discipline or approach. It's a minor issue, but you are setting the reader up in this introduction section so you need to be very clear how your approach is situated with existing work - and where and how it goes beyond it. Previous reviewer reports slightly misunderstood your motivations in places.*

We have removed this metaphor and have made clearer that we are linking socioenvironmental (we prefer this term to socio-ecological as water is abiotic) dynamics with existing IAMs.

*P5 L33 "However, the pathway to capturing these dynamics within global models is not obvious (Müller-Hansen et al., 2017)." this is a citation to a regional study. Verburg et al (2015) may offer a more productive global perspective. But it may not include sufficient discussion on water-energy-climate nexus.*

We have included this useful citation.

*P5 L34 "Thus, the second knowledge gap our framework aims to fill is to incorporate the complex dynamics associated with socioenvironmental systems within the IAM framework, without adding unwanted complexity to those models. We aim to achieve this by focusing on cities." You argue that cities represent a useful aggregation of the agents of interest, that You don't need to capture dynamics of individual farmers or farms for example. This is a centrally important modelling structure decision. It needs to be defended or supported. You provide some high level data that suggests that a*

*focus on cities would represent sufficient capture of the relevant dynamics. Some readers may not be very convinced by that. How else can you effectively argue that resolving agents as cities is the best approach?*

We have expanded on our reasoning for choosing cities as the focus of our study. Page 5, line 27-37

*P8 L1 "However, we recommend tight coupling between dynamic vegetation models such as Lund-Potsdam-Jena managed Land (LPJ-ML) model and a complex hydrological model that captures water fluxes between the soil layer and groundwater reservoir, which is key for computing groundwater recharge rates (Bierkens and van Beek., 2009; Bondeau et al., 2007; Hanasaki et al., 2008b)." It is not clear if you have implemented this method, or have a route towards this method, or if this is an aspiration or some sort of necessary condition for your method to be effective. In any event it is a large piece of work. I think the reader needs to get some understanding of what is involved here.*

We have clarified that this is not yet done, to our knowledge and that it is a major undertaking. We have briefly explained what would be involved in achieving tight coupling between a DGVM and a hydrological model. Page 7 line 38 – Page 8 Line 9

*P9 L7 "The upper level network consists of bilateral trade links among countries. Link weights among countries are calculated using a CGE model, which is coupled to certain IAMs (Stehfest et al., 2013)." More specificity here. Which "certain" IAMs?*

We removed this statement. We discuss how a CGE is linked to the IMAGE IAM on Page 5 Line 1-3.

*P9 L34 "This is a novel and potentially important step in capturing non-equilibrium dynamics within pre-existing equilibrium approaches without adding unwanted complexity to CGEs or IAMs (Farmer and Geanakoplos, 2009)." I appreciate this is an important point. But, if we consider your multi-model framework to be a model (or meta-model), then this represents a very large increase in model complexity. On the one hand you argue that you don't want to add "unwanted complexity" to some kinds of models, on the other your entire approach is about capturing more dynamics and coupling them within a much more complex modelling methodology. Can you be a bit clearer about your motivations here? I think you want to argue that your approach can fit in with the existing modelling 'ecosystem'. You won't be able to take models 'off the shelf' but you don't need to engineer entirely new IAMs. This is a very important point.*

We have clarified this point on Page 10, line 5-11.

*P11 The discussion on how intra city dynamics can impact inter city dynamics - up to the global scale - could be framed in terms of teleconnections. You are arguing that you can only capture such teleconnections in a complex social-ecological system by explicitly resolving agent dynamics and interactions.*

We have made reference to teleconnections in our revised manuscript. Page 11, line 9. Page 4, line 17

*L9 L21 "The main step required to make this framework a reality is the construction of a multi-agent network of city nodes and trade links." I agree! Which makes the lack of detail about such a model quite important. I think we need some framework for this element of the methodology. I would recommend you briefly review the ODD +D protocol. You may find that you are able to able to quickly produce a schematic of your proposed multi-agent model. It's not much more than a list of entities and descriptions, but it will allow you to produce a concise table with which a reader will be able to quickly understand the nature - if not the details - of this central modelling component.*

We have added a table following the ODD +D protocol. Page 11, Line 30. Page 27 and 28

*References*

[revised manuscript text omitted]